# CONTRASTIVE GROUPING-BASED INVARIANT LEARNING FOR GENERALIZABLE GRAPH LEARNING

## ABSTRACT

Recently, Graph Neural Networks (GNNs) have demonstrated remarkable success in various graph learning tasks. However, most existing GNNs fail to generalize under distribution shifts, namely testing and training graphs come from different distributions. Graph invariant learning is proposed to tackle the out-of-distribution (OOD) generalization problem by capturing the invariant relationships between graph features and labels. To this end, most graph invariant learning methods estimate the probabilities of nodes or edges belonging to the invariant subgraphs by measuring these edges' or nodes' contribution degrees of the corresponding edges or nodes to the model's predictive performance. Nonetheless, relying solely on the predictive performance of the model is insufficient to determine whether the given edge or node belongs to an invariant subgraph. To solve this problem, we propose a novel Contrastive Grouping-based Invariant Learning(CGIL) algorithm for OOD generalization on graphs. Our algorithm incorporates the idea of node grouping into the design of learning invariant features. Unlike existing methods that simply employ a mask generator to learn node weights, CGIL tries to cluster graph nodes into an invariant group and several contrast groups. Then CGIL takes the graph connectivity information into account to enforce the graph connectivity inside the invariant group. A contrastive loss constraint is adopted to promote the grouping and invariant subgraph generating procedure. Compared with nine state-of-the-art generalization methods, extensive experiments on four benchmark datasets demonstrate the effectiveness of our proposed CGIL algorithm for the graph classification tasks.

## 1 INTRODUCTION

Graph neural networks (GNNs) have recently gained increasing attention in the domain of graph representation learning and have achieved impressive performance in various tasks on graph data, including social networks Min et al. (2021), molecular graphs Wu et al. (2018), and knowledge graphs Ji et al. (2021). Despite their success, GNN models still face the out-of-distribution (OOD) problem, i.e. the performance of GNN models may significantly degrade when testing graphs have different distributions from training graphs Hu et al. (2020); Wu et al. (2018).

To address the OOD problem, a series of graph learning methods have been proposed Chen et al. (2022); Li et al. (2022a;b). The key idea behind these methods is to learn the invariant subgraphs, of which class labels can stay invariant even if the training and testing graphs have mismatch distributions. To achieve this goal, these graph representation learning methods typically employ an attention module to estimate the probabilities of each node or edge belonging to the invariant subgraphs Brody et al. (2021); Sui et al. (2022). The probability scores of edges or nodes are computed based on the contributions of those nodes and edges to model predictive performance.

However, relying solely on the predictive performance of the model is insufficient to determine whether the given edge or node belongs to an invariant subgraph. In practice, graph data exhibits a widespread presence of shortcut features Knyazev et al. (2019); Fan et al. (2022). The shortcut features usually come from data sampling biases, noisy features, or certain trivial patterns within graphs. These shortcut features are non-causal but discriminative in training data, thereby establishing a spurious correlation with the class label. Consequently, even though the shortcut features

are assigned higher probability weights due to their ability to yield superior predictive performance, they do not actually belong to the set of invariant subgraphs.

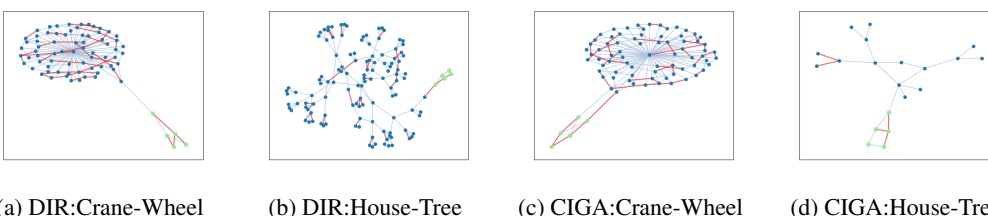



(a) DIR:Crane-Wheel     (b) DIR:House-Tree     (c) CIGA:Crane-Wheel     (d) CIGA:House-Tree



Figure 1: Visualization examples of invariant subgraphs in Spurious-Motif dataset. The learned subgraphs in 1a and 1b are from DIR, and the learned subgraphs in 1c and 1d are from CIGA. The invariant shapes corresponding with labels are Crane and House, and the base shapes are Wheel and Tree. Structures with red color are the invariant subgraphs learned by the corresponding model. Nodes with green color represent the ground truth invariant subgraphs.

As illustrated in Figure 1a ∼ 1d, only a small part of the invariant subgraphs learned by DIR and CIGA are ground truth invariant edges, while most of them contain many shortcut features from the base shape (Wheel and Tree). In this case, the shortcut features contained in the learned invariant subgraphs will reduce the model's generalization ability and prediction accuracy.

In this paper, we focus on designing a new strategy to seek invariant subgraphs and reduce the adverse impact of shortcut features. Specifically, we propose to incorporate the graph connectivity information into the design of the attention module. Different from current attention-based methods that compute one weight of each node and identify an invariant subgraph with these weights, we adopt a soft grouping strategy to compute multiple weights for each node and construct different node groups. Group weights of nodes indicate their probabilities belonging to these groups. Then, a connectivity constraint is introduced to enforce that nodes with higher weights in each group have good connectivity. The node grouping procedure thus overcoming the limitation of the attention module learns attention scores of features solely based on their predictive performance. Any one of these groups is considered as an invariant group, and the others are as contrastive groups. As training progresses, invariant features will gradually converge into the selected invariant group.

By incorporating the idea of soft grouping strategy into the design of learning invariant features, we propose a novel Contrastive Grouping-based Invariant Learning(CGIL), a new algorithm to learn the invariant subgraphs, which can make CGIL generalizable well on data with various distribution shifts. Our CGIL consists of three modules: a node grouper to generate one invariant group and several contrast groups, an invariant subgraph generator to learn potential invariant subgraph from the invariant group, and an invariant subgraph constrainer to constrain the process of node grouping and subgraph generation. In addition, a contrastive loss constraint based on combined invariant subgraphs and groups is proposed to help promote node grouping and invariant subgraph generation.

Specifically, in the node grouper, we jointly optimize the node representation learning and node grouping in an end-to-end way, resulting in one invariant group and several contrast groups. Each group contains all graph nodes, and the weight score of a node reflects the probability that the corresponding node belongs to that group. Then the invariant subgraph generator selects nodes with high weights in the invariant group to generate the potential invariant subgraph. Moreover, in the invariant subgraph constrainer, we use three constraints from different perspectives to constraint the generation of the invariant subgraph: a connectivity constraint to exploit graph connectivity in the graph and classify different groups based on the connected subgraph, a contrastive loss constraint to enforce the node grouping module focus on the invariance of the graph features, and a classification loss to encourage the learned invariant subgraphs to have a strong label prediction ability. Experiments on one synthetic and three real-world datasets with various distribution shifts, show that CGIL can significantly outperform existing methods, demonstrating that our CGIL algorithm has a better generalizable ability than existing methods.

## 2 RELATED WORK

Recently, extensive efforts have been proposed to improve the transparency and generalization capability of graph neural networks (GNNs). According to different learning strategies, existing graph learning methods can be categorized as graph self-supervised learning methods Qiu et al. (2020); You et al. (2020); Yehudai et al. (2021), and graph invariant learning methods Wu et al. (2022); Li et al. (2022b); Chen et al. (2022).

Graph self-supervised learning methods focus on improving model robustness against adversarial attacks during the training procedure. Yehudai et al. (2021) study the ability of GNNs to generalize from small to large graphs, by proposing a self-supervised pretext task that aims at learning useful d-pattern representations. Another representative self-supervised learning method is based on graph contrastive learning. Graph contrastive learning methods GraphCL You et al. (2020) is another representative self-supervised learning method, which aims to obtain augmented graphs from different perspectives, and apply a contrastive learning strategy to maximize their mutual information for training, to enhance the generalizable ability.

Graph invariant learning methods aim to exploit the invariant relationships between the input graph data and labels across distribution shifts while disregarding the variant spurious correlations. Following the recent invariant learning studies Li et al. (2022b), the causes of the label can stay invariant to various distribution shifts.

When generalizing a GNN to graphs with distribution shifts, one of the main challenges is that the environment labels are generally unobserved and highly expensive to obtain. DIR Wu et al. (2022) conducts interventions on variant representations to create multiple interventional distributions, enabling the capturing of invariant rationales while filtering out spurious patterns for robust OOD generalization. However, DIR struggles to find a reasonable ratio to split the invariant rationales and spurious counterparts. In contrast, GSAT You et al. (2020) samples stochastic attention from a parametric Bernoulli distribution to select task-relevant subgraphs and constrain the information flowing from the task-irrelevant graphs to the prediction. GIL Li et al. (2022b) uses a GNN-based subgraph generator to identify the invariant subgraph and defines the rest of the graph as the variant subgraph. To infer the latent environments, GIL adopts an environment inference module to cluster all identified variant subgraphs of the datasets. Recently, CIGA Chen et al. (2022) instantiates the causal features as the critical subgraph that includes the information about the underlying causes of the label, and proposes an information-theoretic objective to extract the desired subgraphs that maximally preserve the invariant intra-class information.

## 3 METHODS

### 3.1 OVERVIEW

Our proposed CGIL framework is illustrated in Figure 2. It is formed by three main components: node grouper, invariant subgraph generator, and invariant subgraph constrainer. Given a graph instance, the node grouper aims to calculate $K$ group weights of each node, including an invariant grouping weight, and cluster graph nodes into one invariant subgraph and $K - 1$ contrast groups. The weights of nodes in the invariant group indicate the probabilities that the corresponding nodes belong to the invariant subgraph. Then, the invariant subgraph generator selects nodes with higher weights to construct the invariant subgraph. Finally, the subgraph constrainer is used to guide the optimization of node grouper and invariant subgraph generator with the goal of ensuring the accuracy of the learned invariant subgraphs. In the following, we will introduce the above three modules in detail, and finally present the training objectives.

### 3.2 NODE GROUPER.

The main purpose of node grouper is to generate an invariant group with node invariant weights. Current methods attempt to compute one weight of each node and identify an invariant subgraph based on these weights. As we discussed in Section 1, with only one weight, these methods cannot leverage the node attributes and graph topology information to find accurately invariant features. To tackle this limitation, we adopt a soft grouping strategy to compute multiple weights for each

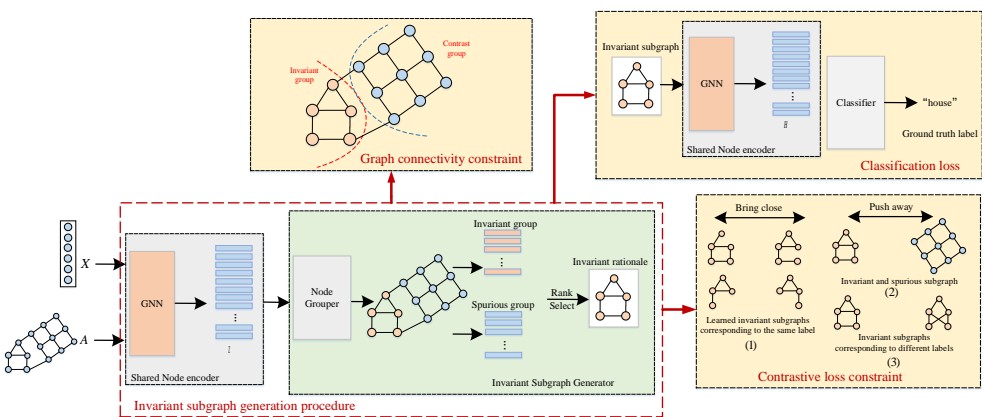

Figure 2: Our proposed CGIL framework

node. Specifically, we generate $K$ groups, $Group_1, Group_2, ..., Group_K$, and each group contains all nodes in a graph instance. Assuming the first group $Group_1$ is the invariant group $Group_{inv}$, the other groups are contrast groups, each node has $K$ group weights indicating their probabilities belonging to the invariant group and other $K - 1$ contrast groups. According to the connectivity constraint introduced in Section 3.4.1, in each group, the soft grouping strategy makes the nodes in a connective subgraph have higher weights than the other nodes and dominate this group. The final aim of the soft grouping strategy is to make each group be dominated by distinct but connective subgraphs. Then by the contrastive loss constraint and classification loss in Section 3.4.2 and 3.4.3, the invariant features will be gathered together in the first group, while the non-invariant features are driven to the other groups.

We define the $j$-th group as follows:

$$
\begin{aligned}
Group_j &= \{Z_1^j, Z_2^j, ..., Z_N^j\} \\
&= \{S_1^j X_1, S_1^j X_1, ..., S_N^j X_N\} \\
&= [S_1^j, S_2^j, ..., S_N^j]^T \cdot [X_1, X_2, ..., X_N] \\
&= (S^j)^T \cdot X
\end{aligned}
\tag{1}
$$

where $Group_j$ contains all $N$ graph nodes, and $Z_i^j$ denotes the representation of $i$-th node in $j$-th group. According to eq.(1), to generate $K$ different node groups, we need to obtain the node representation $X$ and the grouping weight matrix $S$.

In this part, we introduce the generation of node representation and grouping weight matrix. First of all, we adopt a GNN encoder to generate node representation $X$. Given an input graph instance with $N$ nodes as a tuple $G = (\mathcal{V}, \mathcal{E})$, with the node set $\mathcal{V}$ and the edge set $E$. Assuming each node has $D$ features, denoted as $F \in \mathbb{R}^{N \times D}$, and $A \in \{0, 1\}^{N \times N}$ is the adjacency matrix. The GNN encoder employs the following message-passing architecture to integrate neighbor information and generate node embeddings (e.g. GCN Kipf & Welling (2016), GIN Xu et al. (2018)):

$$
H^{(k)} = ReLU(\tilde{D}^{-\frac{1}{2}} \tilde{A} \tilde{D}^{-\frac{1}{2}} H^{(k-1)} W^{(k-1)})
\tag{2}
$$

where $\tilde{A} = A + I$, $\tilde{D}_i = \sum_j \tilde{A}_{i,j}$. $W^{(k-1)}$ is a weight matrix. $H^{(k)}$ is generated from the results of previous message passing $H^{(k-1)}$. The input node embeddings $H^{(0)}$ at the initial message-passing iteration($k = 1$), are initialized using the node features on the graph, $H^{(0)} = F$. After the $k$ steps message passing procedure, the final node embedding is calculated as $X = H^{(k)} \in \mathbb{R}^{n \times d}$. For simplicity, the node encoding procedure can be denoted as $X = GNN(G)$.

Next, we will calculate the grouping weight matrix $S$. $K$ different groups can be seen as labels of $K$ different categories, and the corresponding $K$ grouping weight vectors for each node can be seen as the probability that the node belongs to each group. Therefore, the grouping process can be implemented with a Multi-layer Perceptron(MLP), where the input is the calculated node

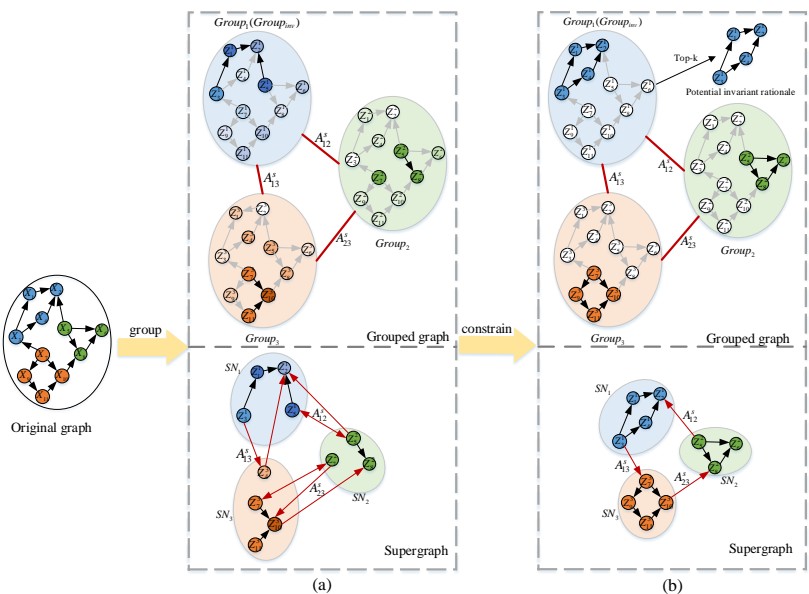

Figure 3: Example of node grouping procedure. The colors of nodes represent their weights for each group. The darker the color of a node, the higher its weight to the group.

embeddings $X$ of $N$ nodes, and the output is the grouping weight matrix $S \in \mathbb{R}^{N \times K}$:

$$S = \sigma(MLP(X); \theta_{MLP}) \tag{3}$$

where $\sigma(\cdot)$ is the softmax function.

Figure 3 is a node grouping example where a graph instance with 11 nodes is clustered into three groups, where $S \in \mathbb{R}^{11 \times 3}$ is the generated grouping weight matrix. Specifically, in Fig.3(a), four nodes with the higher weights in the invariant group are $Z_1^1, Z_2^1, Z_3^1$ and $Z_5^1$, the four nodes are regarded as potential invariant nodes. After end-to-end iterative optimization, the invariant nodes learned by the grouping module in Fig.3(b) are $Z_1^1, Z_2^1, Z_3^1$ and $Z_4^1$, which have better connectivity than those in Fig.3(a).

### 3.3 INVARIANT SUBGRAPH GENERATOR.

In order to further identify nodes with invariant properties, we select the $K$ nodes with the higher contribution degree from the invariant group $G^{inv}$ as potential invariant nodes and construct invariant subgraphs based on these nodes. The contribution of each node in $G^{inv}$ can be determined by the corresponding invariant grouping weight $S^{inv}$. Therefore, we directly sort the invariant grouping weights of the nodes in descending order, use the top-k strategy to select the $K$ nodes with the higher weights, and obtain the corresponding node representations $V_{inv}$:

$$V_{inv} = Top_r(Z) \tag{4}$$

where $r$ is a pre-set sampling ratio(e.g. 40%), we select $K = r \times N$ nodes as potential invariant nodes. Then, we sampled the corresponding adjacency matrix $A_{inv}$ between $K$ nodes from the original graph $G$. Finally, we construct the potential invariant subgraph $G_{inv} = (V_{inv}, A_{inv})$. We can use another GNN encoder to generate invariant node embeddings, and adopt a mean-max pooling to integrate invariant node embeddings into an invariant subgraph representation $SG_{inv}$:

$$SG^{inv} = Pooling(GNN_2(V_{inv})) \tag{5}$$

### 3.4 INVARIANT SUBGRAPH CONSTRAINER.

In the previous sections, we detailed the process of generating a potential invariant subgraph, but we cannot guarantee that the generated invariant subgraph is accurate enough to predict the label

in OOD scenarios. In general, an invariant subgraph has the following three characteristics: 1) Invariant subgraphs usually have strong connectivity. 2) The inter-class distance of invariant subgraph samples in the representation space should be small, while the intra-class distance should be large. 3) Invariant subgraphs should have strong label predictability.

Based on the above characteristics of invariant subgraphs, we constrain the generation process of invariant subgraphs from three different perspectives: connectivity constraint, contrastive loss constraint, and classification loss.

### 3.4.1 CONNECTIVITY CONSTRAINT.

Due to the fact that invariant subgraphs usually have good connectivity, nodes with good connectivity in each group should be assigned with high weights. For this purpose, our connectivity constraint guides the grouping process from three aspects: 1) the nodes with higher weights in the group should have good connectivity; 2) the nodes with higher weights in different groups should not overlap with each other; 3) the numbers of nodes with higher weights in different groups should be similar.

For the first aspect, we consider each group as a supernode and edges among groups as superedges, the original graph $G$ can be summarized into a supergraph $G^s$. We leverage the properties of both supernodes and node groups to make a connective subgraph with higher node weights dominate a group. The representations of supernodes can be denoted as $SN^s = \{SN^{inv}, SN^2, ..., SN^K\}$. The representations of invariant supernode $SN^{inv}$ can be obtained via a pooling operation:

$$SN^{inv} = Pooling(\sum_{i=1}^{N} SN_i^{inv}) = Pooling(\sum_{i=1}^{N} S_i^{inv} \cdot X_i) \tag{6}$$

where $N$ is the total number of graph nodes, and $X_i$ is the generated node embeddings previously. The weight of the superedge is computed as a weighted sum of cross-group edges. Formally, the weight of the superedge between the supernode $m$ and the supernode $n$ is defined as:

$$A_{m,n}^s = \sum_{i=1}^{N} \sum_{j=1}^{N} S_i^m \cdot A_{i,j} \cdot S_j^n \tag{7}$$

Here, $A^s = S^T \tilde{A} S$ represents the weighted adjacency matrix of the supergraph after grouping the nodes, and $S^T \tilde{D} S$ denotes the weighted degree matrix. The sum of the elements on the diagonal of the adjacency matrix $Tr(S^T \tilde{A} S)$ represents the weights of the edges inside the supernodes, while the sum of the elements on the diagonal of the degree matrix $S^T \tilde{D} S$ represents the sum weights of the edges inside and among supernodes. As shown in Fig.2(a), $Tr(S^T \tilde{A} S) = \sum_{i=1}^{3} A_{i,i}^s$ and $Tr(S^T \tilde{D} S) = \sum_{i=1}^{3} D_{i,i}^s$, where $Tr(\cdot)$ represents the trace of the matrix. Obviously, the sum weights of the superedges among different supernodes can be calculated with $Tr(S^T \tilde{D} S) - Tr(S^T \tilde{A} S)$.

In order to make the nodes inside the supernodes as compact as possible, inspired by the Mincut principle Bianchi et al. (2020), we implemented the connectivity constraint to minimize the superedge weights $Tr(S^T \tilde{D} S) - Tr(S^T \tilde{A} S)$ between among supernodes:

$$\min L_{cnt} = \min_{S} -\frac{Tr(S^T \tilde{A} S)}{Tr(S^T \tilde{D} S)} \tag{8}$$

It can be seen that when the value of $L_{cnt}$ is smaller, the weights of the superedges among different supernodes are smaller, and the weights of the edge inside the supernodes are larger. At this time, the inside nodes of the supernodes are more compact and have better connectivity.

In addition, to make the nodes with higher weights in different groups not overlap with each other, we adopt an orthogonality loss term $L_o$ as a supplement to the connectivity constraint. $L_o$ encourages the grouping assignments to be orthogonal and different groups to have the same number of nodes, followed by:

$$\min L_o = \min_{S} \| \frac{S^T S}{\|S^T S\|} - \frac{\mathbf{I}_K}{\sqrt{K}} \|_F \tag{9}$$

where $\|\cdot\|_F$ indicates the Frobenius norm, which helps to encourage $K$ different grouping weights to be orthogonal. And $\mathbf{I}_K = S^T S$ encourages $S$ to assign exactly $N/K$ nodes to each group. During the training procedure, we can find substructures with better connectivity by implementing the connectivity constraint $L_{cnt} + L_o$. However, the found substructures are still not confirmed to be invariant. To address this issue, we further propose a group-based contrastive loss constraint.

### 3.4.2 CONTRASTIVE LOSS CONSTRAINT.

For the invariance of the learned subgraphs, we hope that under different distribution shifts, even if there are small differences in node attributes or graph structures among subgraphs with different labels, there are still large differences in the representation space. For this purpose, we propose a supernode-based contrastive loss constraint suitable for our CGIL. Different from the general contrastive loss constraints based on learned invariant subgraph representations Chen et al. (2022), our CGIL method uses contrastive loss constraint to constrain a concatenated representation, which can be formalized as:

$$G^c = Concat(SG^{inv}; SN^{inv}) \tag{10}$$

where $SG^{inv}$ and $SN^{inv}$ are representations of learned invariant subgraph and supernode. In our proposed CGIL framework, the generation of potential invariant subgraphs is mainly learned from nodes with higher weights inside the invariant supernodes. Meanwhile, the generation of invariant supernodes mainly depends on the node grouping process. Therefore, we hope to guide the soft grouping of nodes while constraining the intra-class compactness and inter-class difference represented by the invariant subgraph. The core idea of our contrastive loss constraint can be expressed as: the learned invariant subgraph corresponding to the same label after grouping is as small as possible in the representation space, while the invariant subgraph corresponding to different labels should be as large as possible in the representation space. Notice that, we do not manually design the soft grouping weight matrix $S$; instead, the node grouper learns $S$ via back-propagation. Having generated learned invariant supernode representations, our contrastive loss constraint guides the generating of $S$ via:

$$\min_S L_{ctr} = \sum_{i=1}^{M} log \frac{e^{\phi(\hat{G}_i^c, \tilde{G}_i^c)}}{e^{\phi(\hat{G}_i^c, \tilde{G}_i^c)} + \sum_{j=1}^{M} e^{\phi(\hat{G}_i^c, \hat{G}_j^c)}} \tag{11}$$

where $M$ is the number of total labels, $\hat{G}_i^c, \tilde{G}_i^c \sim \mathbb{P}_g(G|Y = y)$ are learned invariant supernodes under different distributions that corresponding to the same label $Y = y$. While $\sum_{j=1}^{M} \hat{G}_j^c$ denotes learned invariant supernodes corresponding to the label $Y \neq y$. $\phi(\cdot)$ denotes the similarity function, i.e., cosine similarity. During the training procedure, we implement a contrastive loss constraint to make the similarity distance in the representation space of invariant supernodes corresponding to the same label in the equation as small as possible, while the similarity distance in the representation space of the invariant supernodes with different labels as large as possible.

### 3.4.3 CLASSIFICATION LOSS.

In addition to the above connectivity constraint and contrastive loss constraint, the invariant rationale we learned should also have stable label prediction capabilities. In the OOD task, the learned invariant subgraphs are fed into a classifier to perform label prediction and generate prediction results $\hat{y}$. The classification can be formalized as:

$$\hat{y} = h(SG^{inv})$$
$$L_{cls} = \mathcal{R}(\hat{y}, y) \tag{12}$$

where $SG^{inv}$ is the learned invariant subgraph features previously, $h$ is a classifier, $\hat{y}$ is the predicted result via the classifier $h$, and $y$ is the ground truth label. $\mathcal{R}$ is an empirical risk loss function, such as cross-entropy loss. The goal of classification precision constraint is to constrain the predicted results of the invariant subgraphs learned by the model to be consistent with the ground truth labels.

In summary, by combining Eq.(8) $\sim$ Eq. (12), we obtain the final objectives as shown in Eq. (13):

$$\mathcal{L} = \min_{g,h} \alpha \cdot \mathcal{L}_{ctr} + \beta \cdot \mathcal{L}_{cnt} + \mathcal{L}_o + \mathcal{L}_{cls}$$

where $\alpha$ and $\beta$ represent the weight of the contrastive loss constraint and the connectivity constraint $L_{cnt}$, respectively. $g$ and $h$ respectively represent the parameters of the invariant generator and the classifier, respectively.

**End-to-end Training.** Our proposed CGIL is an end-to-end algorithm. In the training procedure, the node grouper first identifies an invariant group in the graph with good connectivity and strong label prediction ability. Even though the initially generated invariant subgraphs may not be accurate, our subgraph constraint will constrain the grouping process and subgraph generation process from three perspectives: connectivity, invariance, label prediction ability, and update model parameters. As the training building progresses, the model parameters are constantly iterated and updated. The grouping machine will continuously cluster nodes with invariance into our designated invariant groups. Finally, our subgraph generator will select invariant nodes with higher weights from the invariant groups to construct an invariant subgraphs with high connectivity and a strong label prediction ability.

## 4 EXPERIMENTS

### 4.1 EXPERIMENT SETTINGS

In this section, we use one synthetic dataset Spurious-MotifWu et al. (2022), and three real-world datasets: CMNISTArjovsky et al. (2019), Graph-SST5 and Graph-TwitterYuan et al. (2022), details about datasets are illustrated in Appendix A. In different datasets, we use a pre-set hyperparameter $K$ to indicate that we expect to cluster graph nodes into $K$ different target groups, and an early stopping strategy is exploited during training. Here we briefly introduce baseline methods, implementation details, main results, and ablation study. Further hyper-parameter analysis, and visualization analysis are illustrated in Appendix B and Appendix C.

### 4.2 BASELINE METHODS.

We thoroughly compare our CGIL with the following baseline algorithms. Empirical Risk Minimization (ERM) Vapnik (1991) minimizes the average error over multiple domains to learn a robust predictor. Invariant risk minimization (IRM) Arjovsky et al. (2019) is an extension of ERM that regularizes the predictor model to extract invariant features and discard the spurious features. We also compare with several SOTA OOD methods including ASAP Pooling Ranjan et al. (2020), DIR Wu et al. (2022), EIIL Creager et al. (2021), V-REx Krueger et al. (2021), IB-IRM Ahuja et al. (2021), CNC Zhang et al. (2022), and CIGA Chen et al. (2022). Our CGIL follows this class of algorithms and improves the robustness and generalization for GNNs, which helps the models better generalize in out-of-distribution datasets.

### 4.3 IMPLEMENTATION DETAILS

For a fair comparison, CGIL uses the same GNN architectures for GNN encoders as the baseline methods. We set the causal feature ratio as ($r = 0.45$), ($r = 0.6$), ($r = 0.65$), ($r = 0.7$), ($r = 0.4$) for Spurious-Motif, DrugOOD, Graph-SST5, Twitter and CMNIST datasets. We search contrastive loss weight $\alpha$ and MinCut-based clustering loss weight $\beta$ both from $\{0.5, 1, 2, 4, 6, 8, 16, 32\}$ according to the validation performance. During each round of training, we select the best parameters according to the validation performance. For all experiments, each algorithm is repeated 5 times and we report the mean and standard deviation of the classification accuracy.

#### 4.3.1 MAIN RESULTS.

The experimental results of CGIL and its rivals on four datasets are reported in Table 1. The best results are marked in bold. From the experimental results, we can see that, our GIL achieves the top-performing results in 4 datasets. For Spurious-Motif datasets, as bias increases, the strength of the distribution shifts also increases. We can see that in all three biases, including 0.33, 0.6, and 0.9, CGIL outstrips CIGA Chen et al. (2022) averagely by $20\%$ across different degrees of spurious bias. Even if the bias increases to 0.9, The performance of CGIL does not drop significantly. This is due to the graph instance in the Spurious-Motif dataset consisting of one invariant shape and one base shape, thus CGIL can easily cluster graph nodes into an invariant group and a contrast group even under heavy distribution shifts. In the CMNIST, Graph-SST5, and Graph Twitter, CGIL also surpasses CIGA by $3.15\%$, $0.95\%$, and $1.11\%$. These improvements strongly validate that

Table 1: OOD generalization performance on various datasets

| | Spurious-Motif | | | CMNIST-SP | Graph-SST5 | Graph-Twitter |
|---|---|---|---|---|---|---|
| | bias=0.33 | bias=0.60 | bias=0.90 | | | |
| ERM | 54.4±3.50 | 55.48±4.84 | 49.64±4.63 | 13.96±5.48 | 43.89±1.73 | 60.81±2.05 |
| ASAP | 64.87±13.8 | 64.85±10.6 | 57.29±14.5 | 10.23±0.51 | 44.16±1.36 | 60.68±2.10 |
| DIR | 58.73±11.9 | 48.72±14.8 | 41.90±9.39 | 15.50±8.65 | 41.12±1.96 | 59.85±2.98 |
| IRM | 57.15±3.98 | 61.74±1.32 | 45.68±4.88 | 31.58±9.52 | 43.69±1.26 | 63.50±1.23 |
| EIIL | 56.48±2.56 | 60.07±4.47 | 55.79±6.54 | 30.04±10.9 | 42.98±1.03 | 62.76±1.72 |
| IB-IRM | 58.30±6.37 | 54.37±7.35 | 45.14±4.07 | 39.86±10.5 | 40.85±2.08 | 61.26±1.20 |
| CNC | 70.44±2.55 | 66.79±9.42 | 50.25±10.7 | 12.21±3.85 | 42.78±1.53 | 61.03±2.49 |
| CIGA | 77.33±9.13 | 69.29±3.06 | 63.41±7.38 | 44.91±4.31 | 45.25±1.27 | 64.45±1.99 |
| **CGIL** | **93.27±5.70** | **90.63±3.28** | **86.22±7.81** | **48.06±3.63** | **46.20±1.86** | **65.56±1.34** |

CGIL can generalize better under complex distribution shifts. Meanwhile, the low variances on all datasets guarantee the reliability of CGIL in various environments.

### 4.3.2 ABLATION STUDY.

In this section, we conduct ablation studies to verify the effectiveness of different promotion strategies of the proposed CGIL. Firstly, in order to confirm the effectiveness of our grouping module, we merely designed three variant GCIL models: 1) We drop the connectivity and the contrastive loss constraints and denote this version as the Base-model. 2) We implement our CGIL framework without the connectivity constraint, denoted as CGIL-$\mathcal{L}_{cnt}$. 3) We implement our CGIL framework without the contrastive loss constraint, denoted as CGIL-$L_{ctr}$.

Table 2: OOD generalization performance of the variant CGIL models

| Datasets | Spurious-Motif | CMNIST | Graph-SST5 | Graph-Twitter |
|---|---|---|---|---|
| Base-model | 74.87 | 10.31 | 42.63 | 58.91 |
| CGIL-$\mathcal{L}_{cnt}$ | 79.40 | 14.81 | 44.02 | 62.23 |
| CGIL-$\mathcal{L}_{ctr}$ | 78.23 | 31.36 | 45.49 | 62.89 |
| CGIL | 86.22 | 48.06 | 46.20 | 65.56 |

Table 2 shows our ablation experimental results and reveals several insights: (1) CGIL-$L_{cnt}$-$L_{ctr}$ obviously outperforms the Base-model and demonstrates the effectiveness of the base idea of node grouping, which is mainly affected by two reasons. Firstly, the graph instance in Spurious-Motif dataset consists of one base subgraph and one motif, thus can easily cluster into two groups; (3) The results of the CGIL-$L_{cnt}$ are better than the CGIL-$L_{cnt}$-$L_{ctr}$, proving the effectiveness of our proposed connectivity constraint; (4) The results of the CGIL-$L_{ctr}$ outperform the $L_{cnt}$-$L_{ctr}$, prove the effectiveness of our proposed contrastive loss constraint. The result of CGIL attains the best classification performance, further demonstrating the necessity and effectiveness of CGIL, which combines the node grouper and invariant subgraph constrainer.

## 5 CONCLUSION

In this work, we proposed a novel graph invariant learning algorithm CGIL, to study the OOD generalization on graphs via a graph classification task. Our starting point is to integrate node grouping into representation learning. In addition, for existing graph invariant learning methods, we proposed an improved method that combines node attributes and graph connectivity information. Experimental results demonstrate that: 1) By integrating the idea of soft grouping into representation learning, CGIL can learn node representations suitable for graph invariant learning tasks; 2) By considering both node attributes and graph connectivity information, the proposed GIL framework can achieve better generalization capability under various distribution shifts.

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

# A    DATASETS

Table 3 summarizes the datasets used in our paper and introduces the details of dataset partitioning.

Table 3: Details about the datasets used in experiments. The numbers of nodes and edges are taking average among all graphs.

| Datasets | Training | Validation | Testing | Classes | Nodes | Edges |
|---|---|---|---|---|---|---|
| Spurious-Motif | 9000 | 3000 | 3000 | 3 | 44.96 | 65.67 |
| CMNIST | 40000 | 5000 | 15000 | 2 | 56.90 | 373.85 |
| Graph-SST5 | 6090 | 1186 | 2240 | 5 | 19.85 | 37.70 |
| Graph-Twitter | 3238 | 694 | 1509 | 3 | 21.10 | 40.20 |

**Spurious-Motif** is a synthetic dataset from DIR, which involves 15000 graphs, each graph is composed of one base shape (Tree, Ladder, Wheel denoted by $S = 0, 1, 2$ respectively), and one motif shape(Cycle, House, Crane denoted by $C = 0, 1, 2$ respectively). The ground truth label $Y$ is determined by the base shape $C$ solely. Specifically, in the training set, we sample each motif from a uniform distribution, while the distribution of its base is determined by $P(S) = b \times \mathbb{I}(S = C) + \frac{1-b}{2} \times \mathbb{I}(S \neq C)$. We manipulate $b$ to create Spurious-Motif datasets of distinct biases. In the testing set, the motifs and bases are randomly attached to each other.

**Graph-SST5** and **Twitter** are two sentiment sentence classification datasets. Each graph is labeled by its sentence sentiment and consists of nodes representing a word while edges reflect the relationships between different words. The goal is to study explanations that can identify the words with key meanings and the relationships among different words. For SST5, those that have a smaller or equal to the $50^{th}$ percentile average degree are assigned to training, and those that have an average degree larger than the $50^{th}$ percentile while smaller than the $80^{th}$ percentile are assigned to the validation set, and the left are assigned to the test set. For Twitter, we conduct the above split in an inversed order to study the OOD generalization ability for GNNs trained on large-degree graphs to small-degree graphs.

**CMNIST** The goal is to predict a binary label assigned to each image based on the digit. Whereas MNIST images are grayscale, CMNIST colors each image either red or green in a way that correlates strongly with the class label. By eliminating color as a predictive feature, the graph model can result in better generalization.

# B    HYPER-PARAMETER ANALYSIS

In this section, we set a series of experiments to study the effect of three main hyper-parameters in CGIL: the weight of contrastive loss constraint $\alpha$, the weight of contrastive loss constraint $\beta$, and the number of groups $K$. As shown in 4a, we find that when the connectivity constraint $\alpha$ increases in the Spurious-Motif dataset with bias$= 0.33$ and bias$= 0.6$, the classification accuracy shows a trend of initially ascending and then descending as the value of the parameter $\alpha$ increases, while the classification accuracy in bias$= 0.9$ decreases from $\alpha = 2$ to $\alpha = 12$. This phenomenon shows that, as bias increases, CGIL achieves its best performance with a small $\alpha$. In 5a, we can see that the contrastive loss constraint $\beta$ increases in the Spurious-Motif dataset with bias$= 0.33$ and bias$= 0.6$, the classification precision decreases since $\beta = 2$, while the classification precision first increases, then has a slight drop when bias is set as $0.9$. This illustrates that as bias increases, the performance of CGIL can reach its best with a larger $\beta$. 5b shows that in real-world datasets Graph-SST5 and Graph-Twitter, CGIL has stable performance on various group number $K$, and tends to achieve the best results on a larger number of groups $K$.

# C    VISUALIZATION ANALYSIS

We evaluate our model's ability for the OOD graph classification tasks by computing and visualizing the predicted results on the Spurious-Motif dataset. As shown in Fig.5, we visualize the prediction results of the graph samples corresponding to three different label values (circle, house, and crane).

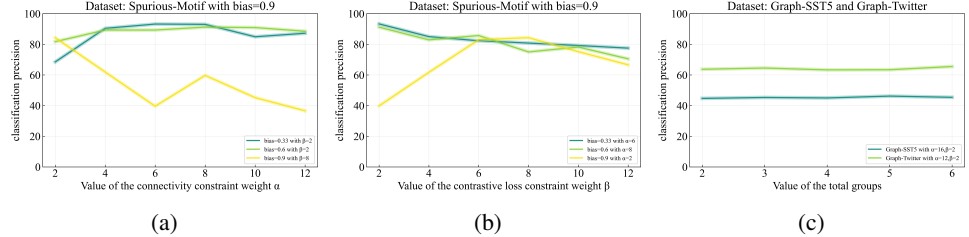

Figure 4: Visualization of CGIL with various hyper-parameters on classification performance

It can be observed that CGIL can learn more accurate invariant subgraphs for model prediction compared to existing methods. Meanwhile, the learned invariant subgraphs contain very few boundary edges. Fig.5a shows a hard example, due to there are two circle shapes in the graph. But in CGIL, we cluster the nodes of two circle shapes into two groups considering the graph connectivity. Then we learn the invariant subgraph from the invariant group in the lower right. This process can verify the reliability of the CGIL framework well.

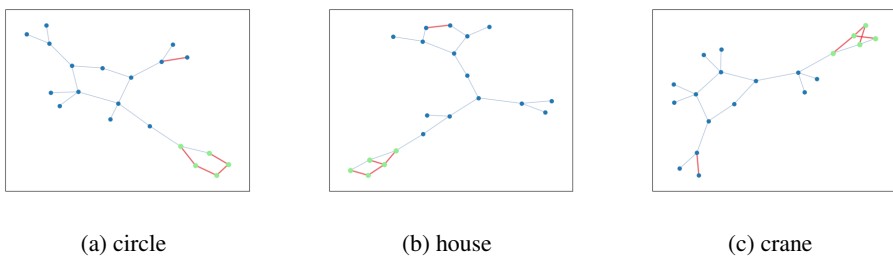

(a) circle       (b) house       (c) crane

Figure 5: Visualization examples of our CGIL on Spurious-Motif datasets

