# OpenReview forum: "Contrastive Grouping-based Invariant Learning for Generalizable Graph Learning"
_ICLR.cc/2024/Conference — Submitted to ICLR 2024_

### Official Review · Reviewer_vit7 · 2023-10-31

**Soundness:** 2 fair
**Presentation:** 2 fair
**Contribution:** 1 poor
**Rating:** 3
**Confidence:** 5

**Summary:**

This article claims that it is difficult for existing invariant graph learning methods to learn invariant subgraphs using label supervision signals, so a new method Contrastive Grouping-based Invariant Learning (CGIL) is proposed to solve these shortcomings and improve the generalization of the GNN model.

**Strengths:**

Experimental results demonstrate the effectiveness of the proposed method.

**Weaknesses:**

I think the main claim of this work is problematic. The authors argue that existing invariant learning methods learn invariant subgraphs directly through data labels, which is obviously wrong.

The reason why invariant learning can capture invariant features is mainly based on the following two assumptions:

1. Training data comes from multiple environments, and model can infer the environment labels from the data distributions.
2. The assumption of graph generation process: graph data is composed of stable (invariant) features and environmental (shortcut) features.
The stable features have two important properties:
- Sufficiency: Stable features contain sufficient information of the label of the target data.
- Invariance: The prediction of stable features remains unchanged in any environments.

The main contribution of this paper is to expose the flaws of these methods, but provide a completely wrong argument.
Why the method proposed in this article can learn invariant subgraphs, the author does not give any reasonable assumptions or theoretical explanations.

**Questions:**

please refer to the Weaknesses.

---

### Official Review · Reviewer_kFr5 · 2023-10-31

**Soundness:** 2 fair
**Presentation:** 3 good
**Contribution:** 1 poor
**Rating:** 3
**Confidence:** 5

**Summary:**

This paper proposes a new graph invariant learning framework, CGIL, for OOD generalization on graphs, which learns to cluster graph nodes into an invariant group and several contrast groups via a connectivity constraint and a contrastive loss constraint. Experiments on four benchmark datasets demonstrate the effectiveness of CGIL.

**Strengths:**

1. OOD generalization is an important problem in graph learning, where exploring graph invariant learning is an interesting topic.
2. The paper is well-organized. The technical details are easy to follow.

**Weaknesses:**

1. The novelty of the proposed framework in this paper is incremental. The method mainly builds on existing techniques, i.e. graph pooling [1] and graph contrastive learning [2]. And improving graph representation learning with clustering and subgraphs is not a novel idea, which has been mentioned in many previous works [3,4]. The authors should refer to, discuss, and compare these works.
2. The proposed framework lacks theoretical support.
3. In realistic scenarios, the invariant relational may be a disconnected subgraph, such as several dispersed functional groups in molecules, which cannot be captured by a single connected subgraph as in the proposed method.
4. The authors only adopted the graph datasets from the synthetic, nlp, and cv domains. Further evaluating the model on several widely used datasets from bioinformatics and social network domains [5] will improve the quality of the paper.

[1] Spectral clustering with graph neural networks for graph pooling. ICML 2020.

[2] Graph contrastive learning with augmentations. NeurIPS 2020.

[3] Clear: Cluster-enhanced contrast for self-supervised graph representation learning. TNNLS 2022.

[4] Multi-Scale Subgraph Contrastive Learning. IJCAI 2023.

[5] Tudataset: A collection of benchmark datasets for learning with graphs. arXiv preprint arXiv:2007.08663.

**Questions:**

See above.

---

### Official Review · Reviewer_bSXr · 2023-11-01

**Soundness:** 3 good
**Presentation:** 2 fair
**Contribution:** 2 fair
**Rating:** 5
**Confidence:** 3

**Summary:**

This paper suggests a GNN framework that given a graph, clusters the nodes into K subgraphs (1 'invariant subgraph' and K-1 'shortcut features'), uses the invariant subgraph for classification, and also uses contrastive learning between subgraphs. This is to increase the performance of the GNNs on out of distribution (OOD) samples. Experiments on 4 OOD generalization datasets and comparison with 8 other methods are conducted with promising results.

**Strengths:**

This paper adds new components to the works of Chen et al. (2022); Li et al. (2022a;b), and I particularly found the application of contrastive learning in this context original. The experimental results indicate that the proposed method outperforms other state of the art methods, which speaks to strengths in quality and significance.

**Weaknesses:**

One major weakness is in clarity and presentation. This is a complicated model with many components, and I found it difficult to follow.

Specific suggestions:
- The authors may consider rewriting or reorganizing the last two paragraphs in Introduction.
- Please provide a table of notations and variables in Appendix.
- (1) does not make sense mathematically. Is Group_j a set or a vector or a scalar?

For others, please see the list of questions below.

**Questions:**

1. I was surprised to see that GSAT and GIL were not included as baseline models in the experiments, since they were discussed in related works. Is there a reason for this choice?

2. Section 4.3 mentions DrugOOD but does not include it in the table. What were those results?

3. Have the authors looked at the model results for Graph-SST5, Twitter and CMNIST beyond accuracy? What kind of subgraphs were found and did they make sense in the context of the original data?

4. There were some components that seemed somewhat unnecessarily complicated. For example, parts of section 3.4.1 looked similar to spectral clustering/graph cut or modularity in community detection. What is the added benefit of these complicated layers ?

---

### Official Review · Reviewer_8ra5 · 2023-11-04

**Soundness:** 3 good
**Presentation:** 3 good
**Contribution:** 2 fair
**Rating:** 1
**Confidence:** 3

**Summary:**

In this paper, to tackle the out-of-distribution (OOD) generalization problem, the authors propose a novel Contrastive Grouping-based Invariant Learning(CGIL) algorithm for OOD generalization on graphs. The proposed algorithm incorporates the idea of node grouping into the design of learning invariant features. Extensive experiments on 4 benchmark datasets demonstrate the effectiveness of the proposed CGIL algorithm for the graph classification tasks.

**Strengths:**

+ Paper is well organized and written.
+ Good experimental results on four datasets.
+ Sound technical backbone.

**Weaknesses:**

- The standard deviation in Table 2 (i.e., ablation study) is missing.
- For Spurious-Motif datasets, I wonder why the authors choose  0.33, 0.6, and 0.9 as biases?
- How to choose the optimal hyperparameters?
- Complexity analysis/running time is missing.
- Code/pseudo code is missing.
- Appendix is missing.

**Questions:**

Please see comments in Weaknesses.

**Details Of Ethics Concerns:**

Not applicable.

---

### Meta-Review · Area_Chair_AVpS · 2023-12-08

**Metareview:**

All the reviewers have shared concerns about the clarity and the presentation of the manuscript. I agree with the reviewers that the manuscript needs to be revised to address the concerns raised. There is no author response to the reviewers' comments. The authors are encouraged to revise the manuscript based on the reviewers' comments.

**Justification For Why Not Higher Score:**

All the reviewers shared significant concerns about the clarity and the presentation of the manuscript and there is no author response to the reviewers' comments.

**Justification For Why Not Lower Score:**

N/A.

---

### Decision · Program_Chairs · 2024-01-16

Reject